# Fatal amyloid formation in a patient's antibody light chain is caused by a single point mutation

Pamina Kazman[1], Marie-Theres Vielberg[1], María Daniela Pulido Cendales[2], Lioba Hunziger[1], Benedikt Weber[1], Ute Hegenbart[3], Martin Zacharias[2], Rolf Köhler[4], Stefan Schönland[3], Michael Groll[1], Johannes Buchner[1]*

[1]Center for Integrated Protein Science Munich at the Department Chemie, Technische Universität München, Garching, Germany; [2]Center for Integrated Protein Science Munich at the Department Physik, Technische Universität München, Garching, Germany; [3]Medical Department V, Amyloidosis Center, University of Heidelberg, Heidelberg, Germany; [4]Institute of Human Genetics, University of Heidelberg, Heidelberg, Germany

**Abstract** In systemic light chain amyloidosis, an overexpressed antibody light chain (LC) forms fibrils which deposit in organs and cause their failure. While it is well-established that mutations in the LC's $V_L$ domain are important prerequisites, the mechanisms which render a patient LC amyloidogenic are ill-defined. In this study, we performed an in-depth analysis of the factors and mutations responsible for the pathogenic transformation of a patient-derived λ LC, by recombinantly expressing variants in *E. coli*. We show that proteolytic cleavage of the patient LC resulting in an isolated $V_L$ domain is essential for fibril formation. Out of 11 mutations in the patient $V_L$, only one, a leucine to valine mutation, is responsible for fibril formation. It disrupts a hydrophobic network rendering the C-terminal segment of $V_L$ more dynamic and decreasing domain stability. Thus, the combination of proteolytic cleavage and the destabilizing mutation trigger conformational changes that turn the LC pathogenic.

*For correspondence:
johannes.buchner@tum.de

Competing interests: The authors declare that no competing interests exist.

## Introduction

Amyloid light chain (AL) amyloidosis is the most common form of systemic amyloidosis. It is the result of an anomalous monoclonal plasma B cell proliferation in the bone marrow which leads to a subsequent overproduction of immunoglobulin light chains (LC) (*Baden et al., 2009*; *Gertz, 2016*). Normally, immunoglobulins (Igs) are secreted as disulfide-bonded complexes comprising two heavy chains (HCs) and two LCs (*Baden et al., 2008*; *Feige et al., 2010*). In AL patients, free LCs excessively escape cellular protein quality control and are secreted into the blood stream. From there, the amyloid precursor LCs can be taken up by cells or deposit extracellularly in organs or tissues (*Baden et al., 2008*; *Falk et al., 1997*; *Feige et al., 2010*; *Gertz, 2016*; *Kastritis and Dimopoulos, 2016*; *Marin-Argany et al., 2016*). The LCs are composed of an N-terminal variable ($V_L$) and a constant domain ($C_L$). The $V_L$ sequence is the result of DNA rearrangements involving a variable (V), a diversity (D) and a joining (J) gene segment (*Bernard et al., 1978*; *Hozumi and Tonegawa, 1976*; *Maki et al., 1980*; *Tonegawa, 1983*). At later stages of differentiation, directed hypermutation of the V region exon further increases the natural diversity of $V_L$ domains (*Feige et al., 2010*; *Jung and Alt, 2004*; *Li et al., 2004*; *Wilson et al., 1998*). As a result, each AL amyloidosis patient possesses a different set of somatic mutations and hence a unique disease-responsible LC sequence (*Blancas-Mejía et al., 2015*). These mutations often destabilize patient LCs compared to non-amyloidogenic LCs. The destabilization was linked to amyloid fibril formation propensity (*Baden et al., 2008*;

**eLife digest** Amyloid light chain amyloidosis, shortened to AL amyloidosis, is a rare and often fatal disease. It is caused by a disorder of the bone marrow. Usually, cells in the bone marrow produce Y-shaped proteins called antibodies to fight infections. In AL amyloidosis, these cells release too much of the short arm of the antibody, known as its light chain, and the light chains also carry mutations. The antibodies are no longer able to assemble properly, and instead misfold and form structures, known as amyloid fibrils. The fibrils build up outside the cells, gradually causing damage to tissues and organs that can lead to life-threatening organ failure.

Due to the rareness of the disease, diagnosis is often overlooked and delayed. People experience widely varying symptoms, depending on the organs affected. Also, given the diversity of antibodies people make, every person with AL amyloidosis has a variety of mutations implicated in their disease. It is thought that mutations in the antibody light chain make it unstable and prone to misfolding, but it remains unclear which specific mutations trigger a cascade of amyloid fibril formation.

Now, Kazman et al. have pinpointed the exact mechanism in one case of the disease. First, tissue biopsies from a woman with advanced AL amyloidosis were analyzed, and the defunct antibody light chain was isolated. Eleven mutations were identified in the antibody light chain, only one of which was found to be responsible for the formation of the harmful fibrils. The next step was to determine how this one small change was so damaging. The experiments showed that after the antibody light chain was cut in two, a process that happens naturally in the body, this single mutation transforms it into a protein capable of causing disease.

In this 'bedside to lab bench' study, Kazman et al. have succeeded in determining the molecular origin of one case of AL amyloidosis. The results have also shown that the instability of antibodies due to mutation does not alone explain the formation of amyloid fibrils in this disease and that the cutting of this protein in two is also important. It is hoped that, in the long run, this work will lead to new diagnostics and treatment options for people with AL amyloidosis.

*Blancas-Mejía et al., 2015*; *Hurle et al., 1994*; *Raffen et al., 1999*; *Ramirez-Alvarado, 2012*; *Wall et al., 1999*). Even though full length LCs and other proteins have recently been found as part of fibrils, patient biopsies predominantly revealed $V_L$ domains that result from proteolytic cleavage of the LC as the main constituent of amyloid deposits (*Blancas-Mejía et al., 2015*; *Buxbaum, 1986*; *Enqvist et al., 2009*; *Gallo et al., 1996*; *Glenner et al., 1970*; *Hurle et al., 1994*; *Nokwe et al., 2014*; *Pepys, 2006*; *Simpson et al., 2009*). Partial unfolding of the pathogenic $V_L$s leads to exposed residue segments that entail the stacking capability of the domain, leading to the tightly packed, dehydrated cross-β amyloid fibril structure (*Annamalai et al., 2016*; *Brumshtein et al., 2018*; *Kelly, 1998*; *Merlini and Bellotti, 2003*). However, changes in stability alone cannot explain the differences in fibril formation propensity (*Nokwe et al., 2014*). Since every patient suffers from a unique LC, it is generally not clear which of the mutated residues trigger the disease. Thus, the underlying reason for $V_L$ variants in AL patients is still largely enigmatic and the mechanistic principles need to be defined.

In this study, we analyzed a patient-derived amyloidogenic LC truncation of the λ subtype with a view to determine the mechanism that renders it pathogenic. The patient variant (referred to as Pat-1) contains 11 mutations in the $V_L$ domain compared to the corresponding germline sequence (referred to as WT-1). A comparison of mutant and germline $V_L$ regarding their structure, stability and amyloid formation propensity allowed us to identify the key mutation responsible for amyloidogenicity and to elucidate its effects on conformational dynamics that result in fibril formation.

## Results

### Origin and sequence of the patient-derived pathological LC

The LC variant studied here was identified in a female patient who was diagnosed with a smoldering myeloma and AL amyloidosis at the age of 50 years at the Amyloidosis Center of the University of Heidelberg. The LC responsible for the disease was of λ subtype. The levels of the free λ LCs were

highly elevated with 7572 mg/l in serum, compared to κ which was normal with 8 mg/l. The clonal plasma cell infiltration of the bone marrow was also high reaching 80%. Heart, kidney, lung, and soft tissue were positive for amyloid deposits and also clinically involved. More specifically, the cardiac stage was IV (*Kumar et al., 2012*) or III b (*Wechalekar et al., 2013*) and the renal stage was 3 (*Palladini et al., 2015*) indicating that the patient was suffering from advanced AL amyloidosis with poor prognosis. Indeed, 4 weeks after initiation of chemotherapy, the patient died of cardiac arrest.

The primary structure of the pathogenic LC, named Pat-1, was deduced from the sequence of the complementary DNA (cDNA) of the plasma cell clone that caused AL amyloidosis. The corresponding germline LC sequence, named WT-1, was identified by database-assisted (abYsis, IgBLAST) primary structure alignments searching for the germline LC with the lowest number of residue changes with respect to the Pat-1 sequence (*Figure 1A*). The two sequences differ in 11 point mutations, all located in the $V_L$ domain. Five of these are located in the constant framework regions (FR) and six are in the hypervariable complementarity determining regions (CDR). We were interested in how frequently the mutant amino acids occur at the respective positions in the antibody repertoire. To this end, we used the Kabat classification system which provides quantitative information on the relative abundance of each amino acid at each position within antibodies (*Johnson and Wu, 2000*; *Wu and Kabat, 1970*). For 9 of the 11 positions, the mutated residues in the Pat-1 $V_L$ are less abundant than the ones present in the WT-1 sequence (*Figure 1—figure supplement 1*) indicating potential negative effects on antibody structure and stability. The two mutated residues that show higher general frequency (S26 and D53) are located in the hypervariable CDRs.

In AL, either the LC, truncated LCs or the $V_L$ domain have been shown to form fibrils (*Blancas-Mejía et al., 2015*; *Buxbaum, 1986*; *Enqvist et al., 2009*; *Hurle et al., 1994*; *Nokwe et al., 2014*; *Pepys, 2006*; *Simpson et al., 2009*). To determine which form is present in the case of Pat-1, we analyzed patient-derived abdominal fat tissue containing amyloid. These deposits were previously shown to be identical to organ-specific fibrils (*Annamalai et al., 2017*). Extraction of the fibrils followed by SDS-PAGE, MALDI fingerprint analysis of the specific band (data not shown) and its mass revealed that in this patient predominantly the $V_L$ domain was deposited (*Figure 1B*, *Figure 1—figure supplement 1*. As the fibril load in organs correlates to the severity of the disease in AL patients, we studied fibril formation in vitro. To test whether proteolytic processing of the LC is a prerequisite for fibril formation, we produced the respective constructs recombinantly in *E. coli* and monitored amyloid formation of the purified proteins. Typically, fibril formation takes a long period of time. It involves unfolding, formation of partially folded intermediates and oligomers which then assemble into fibrils. It was shown earlier that small amounts of SDS presented during the incubation accelerate the process (*Kihara et al., 2005*; *Nokwe et al., 2015*; *Yamamoto et al., 2004*). In this study, we use SDS in concentrations that do not affect the native secondary structure (*Figure 1—figure supplement 2*). In these assays, monitored by thioflavin T (ThT) fluorescence, full length Pat-1 and WT-1 LCs did not form fibrils (*Figure 1C*). However, incubation of the $V_L$ domain of Pat-1 revealed the presence of fibrils after a lag phase, while the WT-1 $V_L$ stayed soluble. We thus conclude that the Pat-1 $V_L$ but not Pat-1 LC is the disease-relevant amyloidogenic species. Transmission electron microscopy (TEM) confirmed the presence of fibrils for Pat-1 $V_L$ samples, whereas no fibrils were detected in the WT-1 $V_L$ and both full length LCs samples (*Figure 1—figure supplement 3*). In line with these findings, all further experiments were performed only with the $V_L$ domains.

## Structural analysis of the Pat-1 and the WT-1 $V_L$ domains

To determine potential structural differences between the two $V_L$ variants Pat-1 and WT-1, the X-ray structure of both proteins were solved to 2.5 Å (Pat-1) and to 1.55 Å (WT-1) resolutions. Despite forming crystals of different space groups, both domains display highly similar structural properties (*Figure 2—figure supplement 1*). They exhibit the typical Ig fold consisting of nine β-sheets forming a greek-key β−barrel topology with the three hypervariable CDR loops in spatial proximity (*Figure 2A*). At first sight, none of the mutations is located at a position that could explain the difference in fibril formation tendency. Six out of the 11 variations are found in the CDRs (T26S, S28N, V30F, G32D, E53D and S55D). The five substitutions in the FRs are located in two regions. Two of them which flank the CDR3 loop are in close proximity to the $V_L$-$V_H$ interface (Y90F and T105S), without disrupting the contact surface in comparison to WT-1. The remaining three mutations (P15L, L81V and Q82L) are opposite to the antigen-binding site, located close to the C-terminal part of the $V_L$ domain. Thus, judging from the positions of the mutations in the domain structure and the

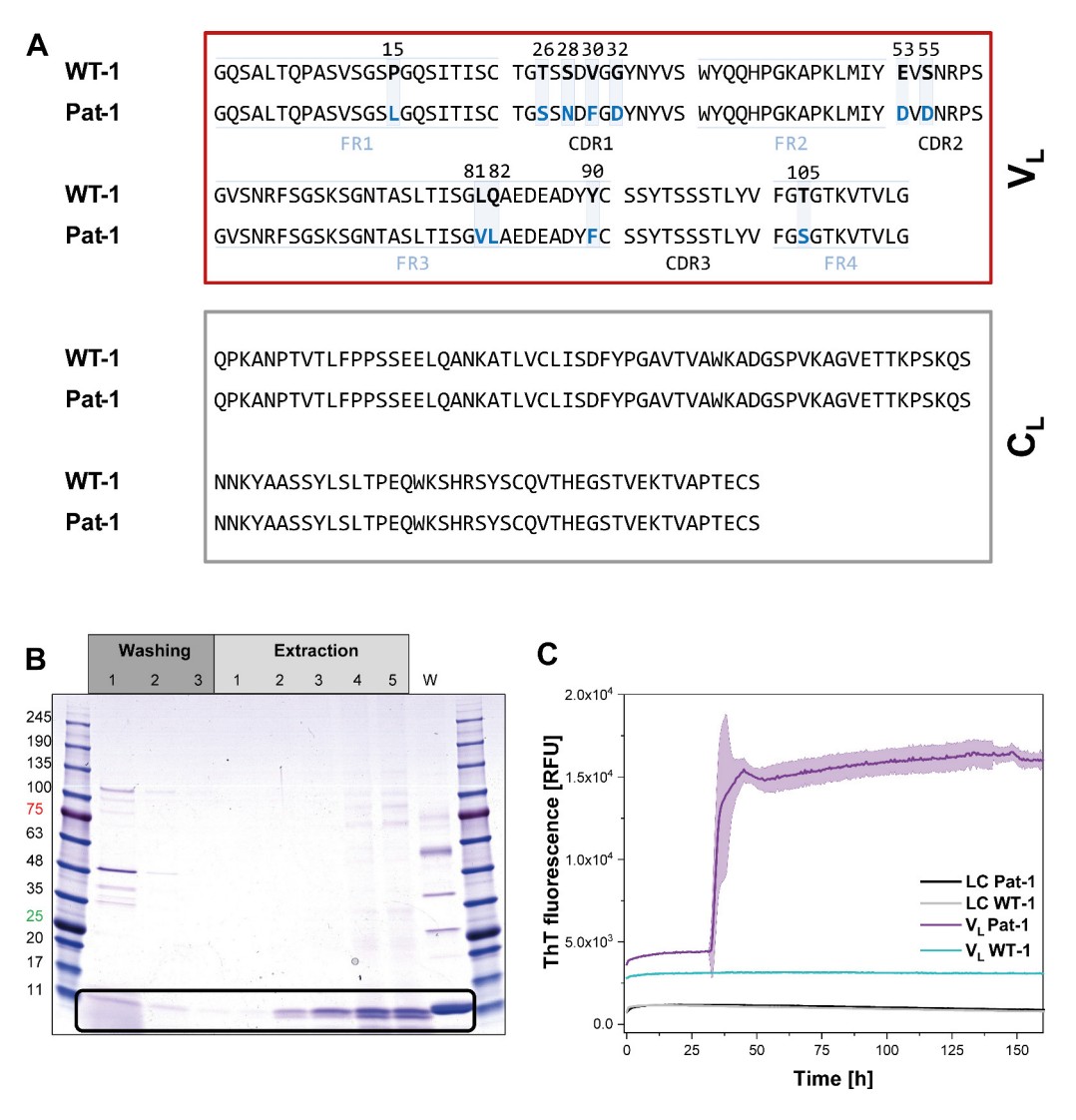

**Figure 1.** Identification of the $V_L$ domain as predominant fragment for fibrillation in the patient. (**A**) Sequence alignment of the patient-derived Pat-1 LC with the corresponding germline WT-1. The Pat-1 sequence was obtained cDNA sequencing of the disease-causing bone marrow plasma cells, the WT-1 sequence was obtained by sequence comparison in Ig BLAST and abYsis. The $V_L$ and $C_L$ domain is surrounded by a red and gray box, respectively. The $V_L$'s FRs and, CDRs were defined by Ig BLAST. The 11 point mutations are depicted in blue boxes. (**B**) Fibril extraction from abdominal fat tissue of the patient revealed the predominant deposition of LC truncations containing mostly the $V_L$ domain in the patient's fibrils. The black box indicates the band of the $V_L$ domain at ~11 kDa. The first three steps correspond to washing steps, the next five to extraction steps, the last lane named 'W' corresponds to an initial tissue wash. (**C**) ThT fluorescence of the LC and $V_L$ domains of Pat-1 and WT-1 monitored over time. An increase in fluorescence indicates fibril formation. The assay was performed in PBS buffer containing 0.5 mM SDS at 37°C and shaking. The continuous lines show the mean value of triplicates with SD as transparent coloured background.

The online version of this article includes the following source data and figure supplement(s) for figure 1:

**Figure supplement 1.** Analysis of sequence mutations and sequence truncation.

**Figure supplement 1—source data 1.** Frequencies of amino acids at the specific positions in WT-1 and Pat-1 according to the Kabat numbering scheme.

**Figure supplement 2.** Comparison of far UV CD-spectra of Pat-1 and WT-1 with and without 0.5 mM SDS.

**Figure supplement 3.** TEM micrographs of $V_L$s of (**A**) Pat-1 and (**B**) WT-1 and LCs of (**C**) Pat-1 and (**D**) WT-1.

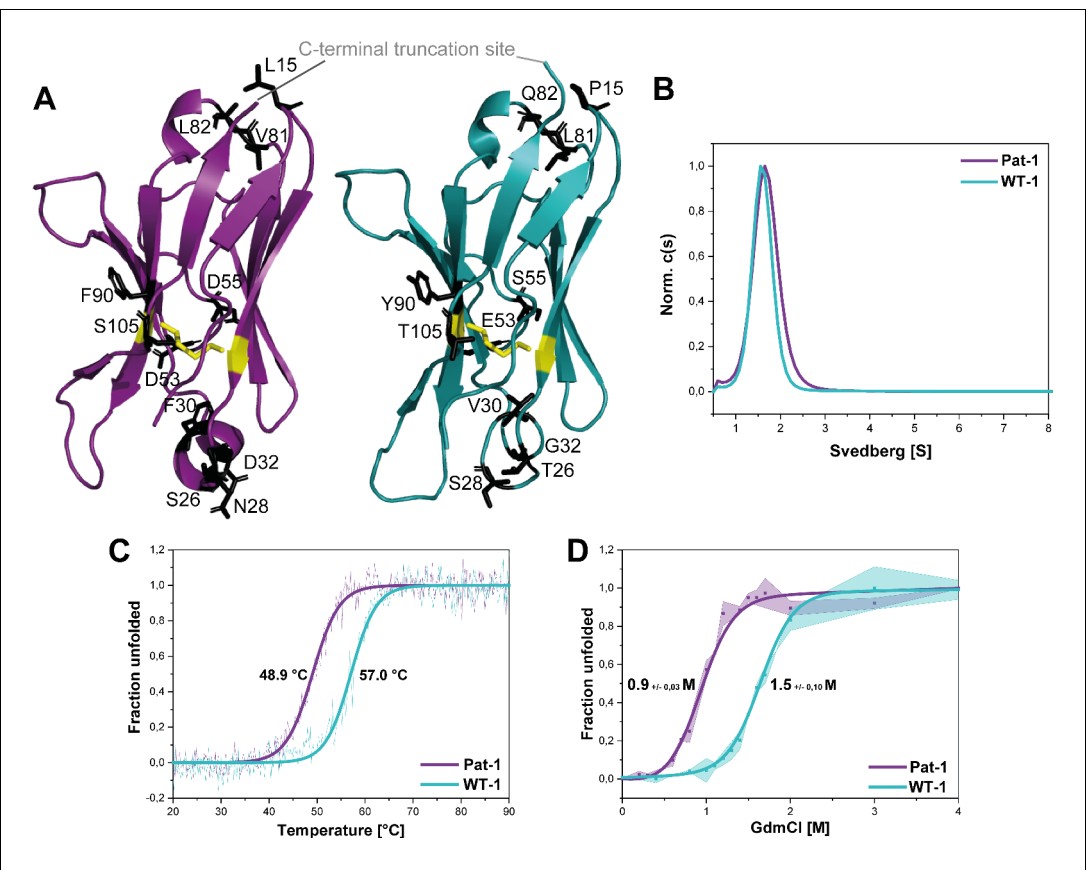

**Figure 2.** Structural characterization and stability comparison of the VL domains Pat-1 and WT-1. The pathogenic $V_L$ Pat-1 is highlighted in purple, the germline $V_L$ in cyan. (**A**) Crystal structure of the $V_L$s (PDB IDs: 6SM1, 6SM2). The intramolecular disulfide bond is shown as yellow sticks. The nine point mutations are depicted as black sticks. (**B**) Analytical ultracentrifugation sedimentation velocity runs with Svedberg values of ~1.6 s for both $V_L$s. The data were analyzed using the continuous c(S) distribution mode of SEDFIT. (**C**) Temperature-induced unfolding transitions of Pat-1 and WT-1. The dots show the raw data, the continuous line shows the theoretical curves derived by fitting the data to a Boltzmann function. (**D**) GdmCl-induced unfolding transitions of both Pat-1 and WT-1 monitored by tryptophan fluorescence at 349 nm. The measurements were performed three times and the continuous lines shows the mean data fit to a two-state unfolding mechanism with SD. The significance corresponds to a p value of p=0.000996. Source files of the crystallographic data collection and refinement statistics are available in *Figure 2—source data 1*.

The online version of this article includes the following source data and figure supplement(s) for figure 2:

**Source data 1.** Crystallographic data collection and refinement statistics for WT-1 and Pat-1.
**Figure supplement 1.** Superposition of the crystal structures of Pat-1 and WT-1.

superposition of both crystal structures (*Figure 2—figure supplement 1*), no reasonable conclusions concerning their contributions to fibril formation could be drawn.

## The $V_L$ domains differ in thermodynamic stability and fibril formation propensity

To gain further insight into the consequences of the mutations on the conformation of the $V_L$ domain, we analyzed whether the variants forms dimers in solution as reported for some LCs and their fragments (*Brumshtein et al., 2014*; *Rennella et al., 2019*). Analytical ultracentrifugation showed that the $V_L$s studied are monomers in solution with sedimentation coefficients of approximately ~1.6 s (*Figure 2B*).

To test whether the mutations affect the thermal stability of the Pat-1 and WT-1 $V_L$, the loss of secondary structure upon heating was monitored by far-UV CD-spectroscopy (*Figure 2C*). The melting temperatures ($T_m$) at which 50% of the protein is unfolded were determined to be 48.9°C and

57.0°C for Pat-1 and WT-1, respectively. Furthermore, the stabilities against chemical unfolding were followed by changes in the tryptophan fluorescence intensity in the presence of increasing GdmCl concentrations. Both $V_L$ domains showed sigmoidal unfolding curves and the $c_m$-value (the GdmCl concentrations at which 50% of the protein is unfolded) was found to be increased by 0.46 M GdmCl for WT-1 compared to Pat-1 (*Figure 2D*). These results revealed that the patient-derived variant is significantly less stable than the germline protein.

### The mutation V81L strongly affects the stability and fibril formation propensity of Pat-1

To identify the mutation(s) decisive for stability and amyloidogenic properties of the Pat-1 $V_L$ domain, we substituted all mutations of Pat-1 individually with the respective residue present in the WT-1 $V_L$. When we analyzed their thermal and chemical stabilities, we found that 10 out of the 11 Pat-1 $V_L$ point mutants either (i) only had minor effects on the stability (S26T, N28S and S105T), (ii) were even destabilizing the domain (L15P, F30V and D53E), (iii) showed a negative effect on thermal stability (D32G and D55S), or, iv) showed a slight increase in chemical stability (L82Q and F90Y) (*Figure 3A,B*). In contrast, the mutation V81L resulted in a substantial shift of the Pat-1 $V_L$ towards the thermal and chemical stability of WT-1 with a $T_M$ of 54.6°C and a $c_M$ of 1.36 M (WT-1: $T_M$ = 57.0°C, $c_M$ = 1.43 M) (*Figure 3A,B*).

Remarkably, when we performed fibril formation assays, we found that all Pat-1 point mutants except V81L formed amyloids (*Figure 3C*), suggesting that valine at position 81 is the main cause for the pathogenicity of Pat-1. All fibril forming variants showed an even lower halftime ($t_{1/2}$) of fibril formation than Pat-1. The maximal ThT intensity, which seems to correlate to the amount of fibrils present, was in most cases equal to Pat-1. The only exception is the F30V variant which showed an increased maximum ThT fluorescence (*Figure 3—figure supplement 2*).

Valine and leucine are closely related amino acids; both exhibit hydrophobic characteristics and differ in only a methylene moiety. Since isoleucine is identical to leucine concerning the constituent atoms, we substituted V81 by isoleucine and analyzed this mutant regarding stability and amyloid formation propensity. Intriguingly, V81I did not exhibit a significant increase in thermal stability compared to Pat-1 (*Figure 3—figure supplement 1A*). Furthermore, we found that V81I, unlike V81L, readily forms fibrils (*Figure 3—figure supplement 1B*). Thus, we conclude that the protective effect observed for the V81L variant is highly specific to leucine. While the experiments so far suggest that the V81L mutation can rescue the Pat-1 $V_L$ from amyloid formation, it was an open question whether vice versa replacement of leucine by valine in the context of WT-1 would be sufficient to induce fibril formation. We therefore cloned WT-1 L81V and analyzed its properties. The thermal stability of WT-1 L81V decreased by 5.5°C compared to WT-1 (*Figure 3D*) and the $c_M$ value for chemical unfolding, is reduced by 0.42 M GdmCl (*Figure 3E*). Hence, the introduction of valine at position 81 shifts WT-1 toward the stability of Pat-1. In the same line, upon incubation the WT-1 L81V mutant showed an increase of ThT fluorescence indicating fibril formation (*Figure 3F*). Together, these results show that it is possible to reconstitute the properties of the AL patient mutant by exchanging only one amino acid in the wild-type protein.

### Valine 81 is located in a surface-exposed hydrophobic spot crucial for stability and amyloidogenicity

The crystal structure of Pat-1 showed that V81 is in close proximity to L15 and L82 opposite of the antigen-binding site (*Figure 2A*). Focusing on the surface properties in this region, we found that these three amino acids form a hydrophobic surface patch which is not present in the WT-1 protein (*Figure 4A,B*). To determine the combined effect of the mutations at position 15, 81 and 82 (L15P, V81L and L82Q), we designed additional double and triple mutants carrying all possible combinations of these substitutions in Pat-1 and determined their stabilities against thermal and chemical denaturation. The combination of the substitutions at position 81 and 82 enhanced the increase in stability slightly, whereas mutating leucine at position 15 to proline in combination with the other point mutations led to a decrease in stability (*Figure 4C,D*). Fibril formation kinetics monitored by ThT fluorescence showed that the stability changes of the analyzed $V_L$ variants correlate with the amyloid propensity. The anti-amyloidogenic effect of V81L could be preserved in combination with the L82Q mutation. The presence of L15P in the mutants increased fibril formation (*Figure 4E*,

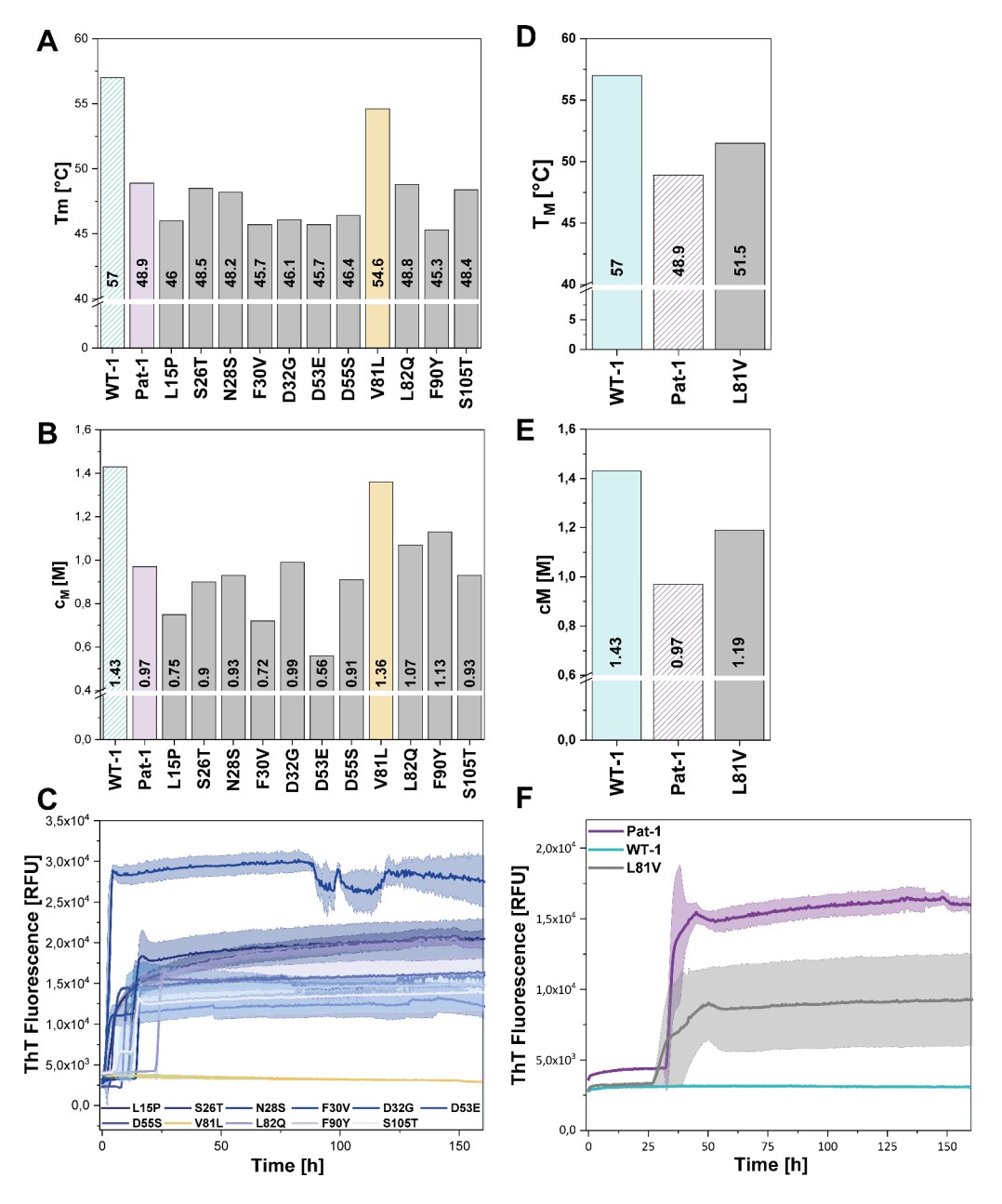

**Figure 3.** Impact of point mutations on the stability of Pat-1 and WT-1. (**A**) Comparison of $T_m$ and (**B**) $c_M$ values of Pat-1, WT-1 and single point mutants of Pat-1. The $T_m$ values were derived from fitting the thermal-induced unfolding transition data to a Boltzmann function. $c_M$ values were obtained by fitting the chemical unfolding data according to a two-state unfolding model. (**C**) Fibril formation assay of single-point mutants followed by ThT fluorescence. (**D**) Comparison of $T_m$ and (**E**) $c_M$ values of Pat-1, WT-1 and the single point mutant L81V. The $T_m$ values were derived from fitting the thermal-induced unfolding transition data to a Boltzmann function. $c_M$ values were obtained by fitting the chemical unfolding data according to a two-state unfolding model. (**F**) Fibril formation assay followed by ThT fluorescence. The assay was performed in PBS buffer containing 0.5 mM SDS at 37°C and shaking. The continuous lines show the mean value of triplicates with SD as transparent coloured areas.

The online version of this article includes the following figure supplement(s) for figure 3:

**Figure supplement 1.** Impact of the V81I substitution on the thermal stability and fibril formation propensity of Pat-1.

**Figure supplement 2.** Comparison of $t_{1/2}$ and maximal ThT fluorescence values of the $V_L$ variants.

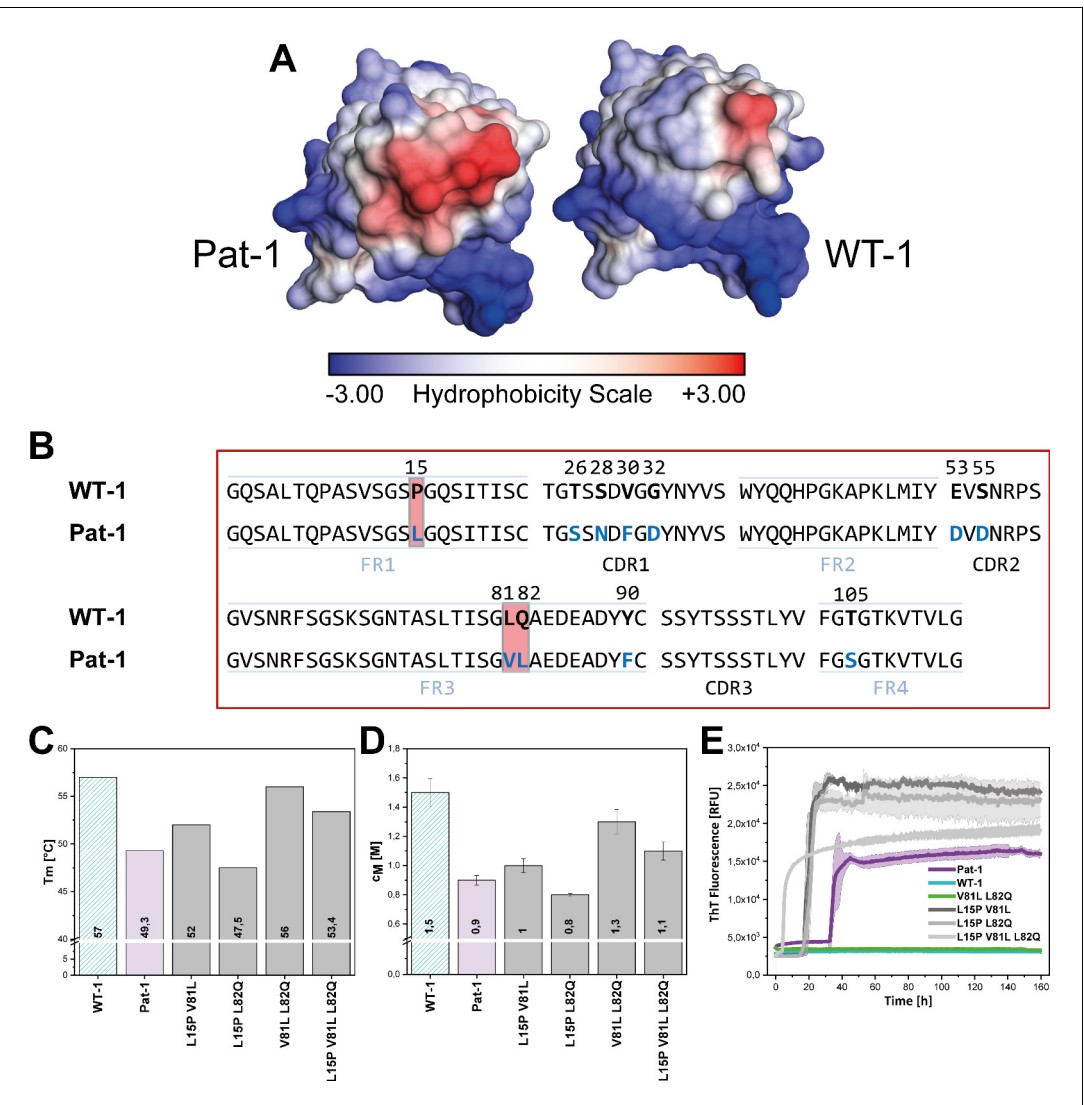

**Figure 4.** Influence of amino acid substitutions altering the hydrophobic of Pat-1. (**A**) Surface representation of the crystal structures of Pat-1 and WT-1. The surface amino acids are colored according to their hydrophobicity. (**B**) Sequences of Pat-1 and WT-1 with the amino acids located in the hydrophobic surface patch of the $V_L$ domains highlighted in red. (**C**) Thermal and (**D**) chemical stabilities of WT-1, Pat-1 and the single, double and triple point mutants varying the hydrophobic surface of Pat-1. Thermal stabilities were obtained by fitting the thermal unfolding data to a Boltzmann fit. Chemical stabilities were obtained by fitting GdmCl unfolding data according to a two-state unfolding model. The measurement was performed three times and the mean was taken. (**E**) Fibril formation was followed by ThT fluorescence. The assay was performed in PBS buffer containing 0.5 mM SDS at 37° C and shaking. The continuous lines show the mean value of triplicates with SD as transparent colored areas. The online version of this article includes the following figure supplement(s) for figure 4:

**Figure supplement 1.** TEM micrographs of hydrophobic mutants of Pat-1.

---

*Figure 4—figure supplement 1*). From these results, we conclude that the hydrophobic surface spot in Pat-1 plays a key role in pathogenesis. Minimizing the area by the back-mutation of the hydrophobic residue L82 to glutamine supports the anti-amyloidogenic effect of V81L. L15P alone and in all combinations tested had negative effects on stability and fibril formation.

## Altered hydrophobic interactions lead to higher conformational dynamics of the patient $V_L$ domain

To gain insight into conformational dynamics of the Pat-1 and WT-1 $V_L$ domains, we monitored the kinetics of hydrogen/deuterium exchange in the protein backbone by mass spectrometry (H/DX-MS) (*Bai and Englander, 1996*; *Rand et al., 2014*). We obtained complete sequence coverage in the H/DX-MS for both variants. Our analysis indicates that Pat-1 exhibits higher conformational dynamics as indicated by an overall higher HD/X rate (*Figure 5A*). The most affected segment was the C-terminal framework region 4 (FR4, residues 101–113). Additionally, the region from amino acid 11 to 20 (FR1), comprising the residue at position 15 which is part of the surface-exposed hydrophobic spot, is striking since the fractional uptake of deuterium in Pat-1 increases in this area. When the differences in fractional deuterium uptake between Pat-1 and WT-1 are mapped onto the X-ray structure of Pat-1, it becomes obvious that the Pat-1 $V_L$ shows a higher flexibility in most parts of the domain compared to WT-1 (*Figure 5B*).

To determine the underlying molecular reasons for the differences in dynamics and stability, we analyzed side chain interactions in the structures of the $V_L$ domains using the BIOVA Discovery Studio software. We found that the mutations at position 15, 81 and 82 result in a change in hydrophobic interactions between Pat-1 and WT-1 (*Figure 6*). P15 is located in FR1, in a region that shows a conspicuous change in deuterium exchange. In WT-1, this proline is not involved in a hydrophobic interaction network. In contrast, in the patient mutant L15 interacts with three amino acids located in FR3, including the two other point mutations in the hydrophobic surface spot, and two residues in

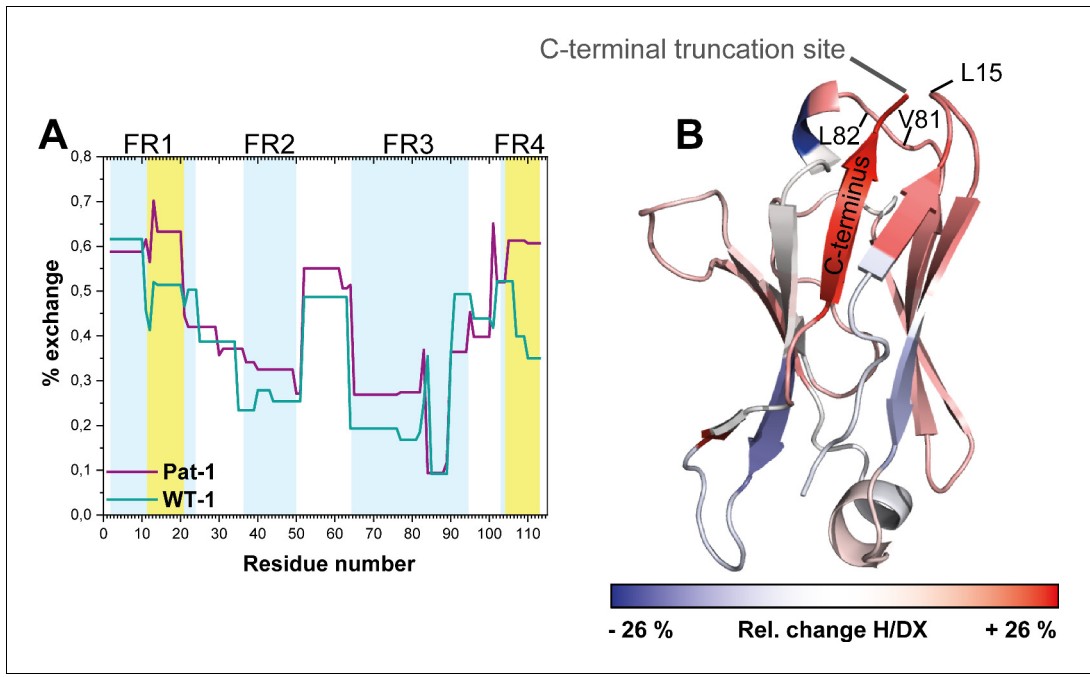

**Figure 5.** Comparison of hydrogen-deuterium exchange rates of Pat-1 and WT-1. (**A**) The magnitude of exchange of Pat-1 is shown in purple, for WT-1 in cyan. FRs are highlighted in light blue, CDRs in white. Pat-1 shows overall higher exchange rate. Strongly affected regions are highlighted in yellow. The Experiment was performed twice. (**B**) The relative change in fractional uptake is mapped on the Pat-1 crystal structure. Negative and positive relative change values in H/DX indicate more dynamic WT-1 and Pat-1 regions, respectively. Relative change of H/DX of Pat-1 segments with respect to WT-1 is color coded from blue (low) to red (high). Source files for the H/DX measurement sets are available in *Figure 5—source datas 1–4*.

The online version of this article includes the following source data for figure 5:

**Source data 1.** First set of HD/X values for WT-1.
**Source data 2.** Second set of HD/X values for WT-1.
**Source data 3.** Figure 5—source data 3. First set of HD/X values for Pat-1.
**Source data 4.** Second set of HD/X values for Pat-1.

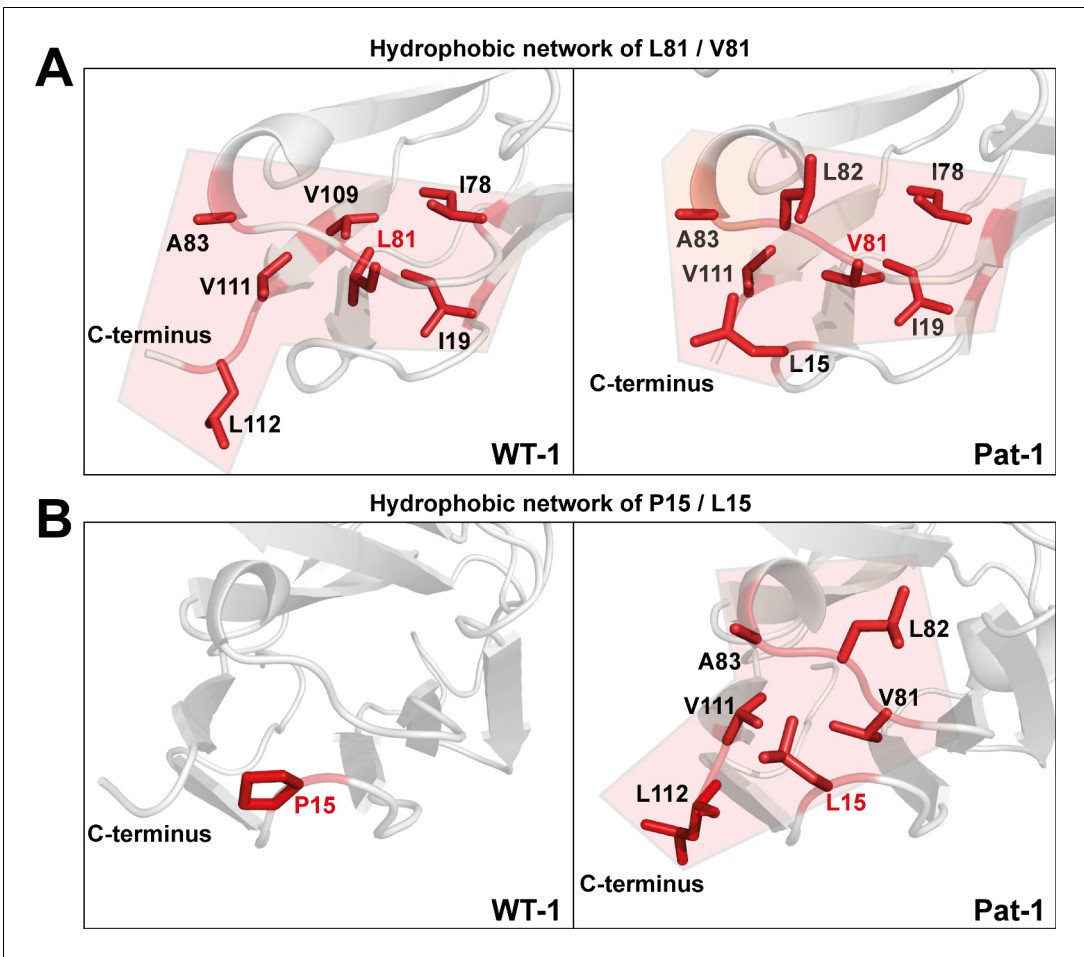

**Figure 6.** Hydrophobic networks of amino acids at position (**A**) 81 and (**B**) 15. The amino acids included in the hydrophobic network are shown in red stick representation. The area of the network is highlighted in light red. Source file of the hydrophobic interaction patterns is available in *Figure 6—source data 1*.

The online version of this article includes the following source data for figure 6:

**Source data 1.** Hydrophobic interaction pattern of the three mutated amino acids at positions 15, 81 and 82, located in the altered hydrophobic surface area of the V$_L$ variants Pat-1 and WT-1.

the C-terminal region (V111 and L112). Thus, the contact of L15 to the very dynamic C-terminal segment, as indicated by the HD/X rate, induces the enhanced conformational dynamics of the respective loop region in FR1. V81 in Pat-1 forms six hydrophobic interactions with residues in FR2 and FR3, two of them, the L15 and L82 point mutations, are located within this hydrophobic surface area. However, V81 contacts only one amino acid in the C-terminal segment (V111). The hydrophobic interaction pattern of the WT-1 L81 misses one interaction each in FR2 and in FR3 but two more interactions to the C-terminal region appear. Overall, this leads to three hydrophobic interactions of L81 to the C-terminal region (V109, V111 and L112). For the mutation at position 82, the biggest effect comes from the change of the nature of interaction from hydrophobic to hydrophilic and the resulting decrease of hydrophobic surface area. L82 can form five hydrophobic interactions with FR2-4. Q82 however forms hydrophilic interactions to four residues, all located in FR3: R64, E84, D85 and E86. We conclude that the network of interactions in the C-terminal region is markedly changed in Pat-1 by the mutations at position 15, 81 and 82 leading to a highly dynamic C-terminal region.

## Modified hydrophobic interaction pattern favors dissociation of the C-terminal region

To further analyze the effect of the protective back mutation of V81L in Pat-1 and the destructive substitution of L15P on the C-terminal network, we performed molecular dynamics (MD) free energy simulations. By Umbrella Sampling (US) simulations, the dissociation of the C-terminal segment (residues beyond 102) was induced during MD-simulations for WT-1, Pat-1, Pat-1 V81L and the double mutant Pat-1 V81L L15P and the associated change in free energy was calculated. The free energy of dissociating this part of the structure from the otherwise folded protein can be taken as a relative estimate for the influence of the mutation on the stability of the protein fold. The calculated free energy profiles of the dissociation of the C-terminal region are shown in *Figure 7*, RMSD values are shown in *Figure 7—figure supplement 1*. The free energy simulations exhibit similar dissociation free energies for the WT-1 and the single back mutation V81L, which were higher (by ~1 kcal·mol$^{-1}$) in comparison to Pat-1 and the double back mutation V81L L15P. This indicates an increased protein stability for WT-1 and the back mutation of Pat-1 V81L. The effect of the V81L mutation is mainly due to a small cavity that is present in the case of V81 and that is filled if V81 is replaced by a slightly larger residue (V81L) (*Figure 7*). However, a destabilizing effect due to the additional back mutation L15P could be observed. Here, the double back mutation exhibited an opposite effect compared to the single back mutation, by producing a significant reduction of the dissociation free energy (by ~1 kcal·mol$^{-1}$) relative to the patient mutant. This effect can be explained by a change in the local protein backbone structure (due to the L15P) that changes the local packing geometry and offsets the effect of the V81L mutation.

## Discussion

Patients suffering from systemic AL amyloidosis often show different patterns of affected organs as well as widely varying symptoms. This impedes both diagnosis and treatment (*Blancas-Mejía et al., 2015*; *Gertz, 2016*; *Gertz and Kyle, 1997*; *Hurle et al., 1994*; *Li et al., 2004*; *Merlini and Bellotti, 2003*; *Palladini and Merlini, 2009*; *Ramirez-Alvarado, 2012*; *Schönland et al., 2012*). Even though the importance of mutations in the V$_L$ domain of overexpressed LCs is established (*Baden et al., 2009*; *Hurle et al., 1994*; *Martin and Ramirez-Alvarado, 2010*; *Nokwe et al., 2014*), the mechanisms of pathogenicity are still unclear. The molecular analysis is hampered by the presence of various mutations in V$_L$ domains from different patients and the uniqueness of each patient's LC. Also, the importance of the proteolytic cleavage for amyloidogenicity is in most cases unclear. Our comprehensive characterization of a pathogenic LC variant allows us to answer these questions and provides a comprehensive picture of the requirements to initiate fibril formation.

For Pat-1, the full length LC is resistant to fibril formation. Proteolytic cleavage in the linker region between the V$_L$ and C$_L$ domain is a prerequisite to unleash the amyloidogenic potential of the mutations in V$_L$. The deposition of truncations of very similar sizes in the patient's tissue suggests a complex picture concerning the proteases involved in the generation of pathogenic LC truncations. Recent cryo-EM structures of AL fibrils showed a highly diffuse density for the C-terminal residues of the V$_L$ domain indicating a flexible orientation. Thus, the exact pathogenic truncation site between V$_L$ and C$_L$ does not seem to be crucial (*Radamaker et al., 2019*; *Swuec et al., 2019*). NMR studies further suggest the existence of different fibril topologies in V$_L$ amyloid fibrils. They may contain well-ordered and rigid C-terminal ends or a highly ordered hydrophobic core domain (*Hora et al., 2017*; *Piehl et al., 2017*). For a better understanding of the disease, we conclude that the analysis of biopsies will provide important information about the presence of full length or truncated LCs in the patient's organs and tissues and thus about the disease-causing LC species (*Annamalai et al., 2016*; *Ramirez-Alvarado, 2012*; *Weber et al., 2018*).

It is obvious that in AL, the large amount of insoluble aggregates deposited in organs interferes with their function. However, the mechanism of toxicity remains unknown. In general, there are different hypotheses on the nature of the toxic species in amyloid diseases. Besides fibrils, Oligomers formed on the pathway to highly organized amyloid structures are considered toxic (*Merlini, 2017*; *Riek and Eisenberg, 2016*). In this study, we focused on analyzing fibril formation as their presence is directly correlated to pathogenicity in AL.

The results of the in vitro fibril formation assays are in line with the predominant deposition of V$_L$ species in the patient: neither the Pat-1 LC nor the WT-1 LC formed fibrils; also the WT-1 V$_L$ was

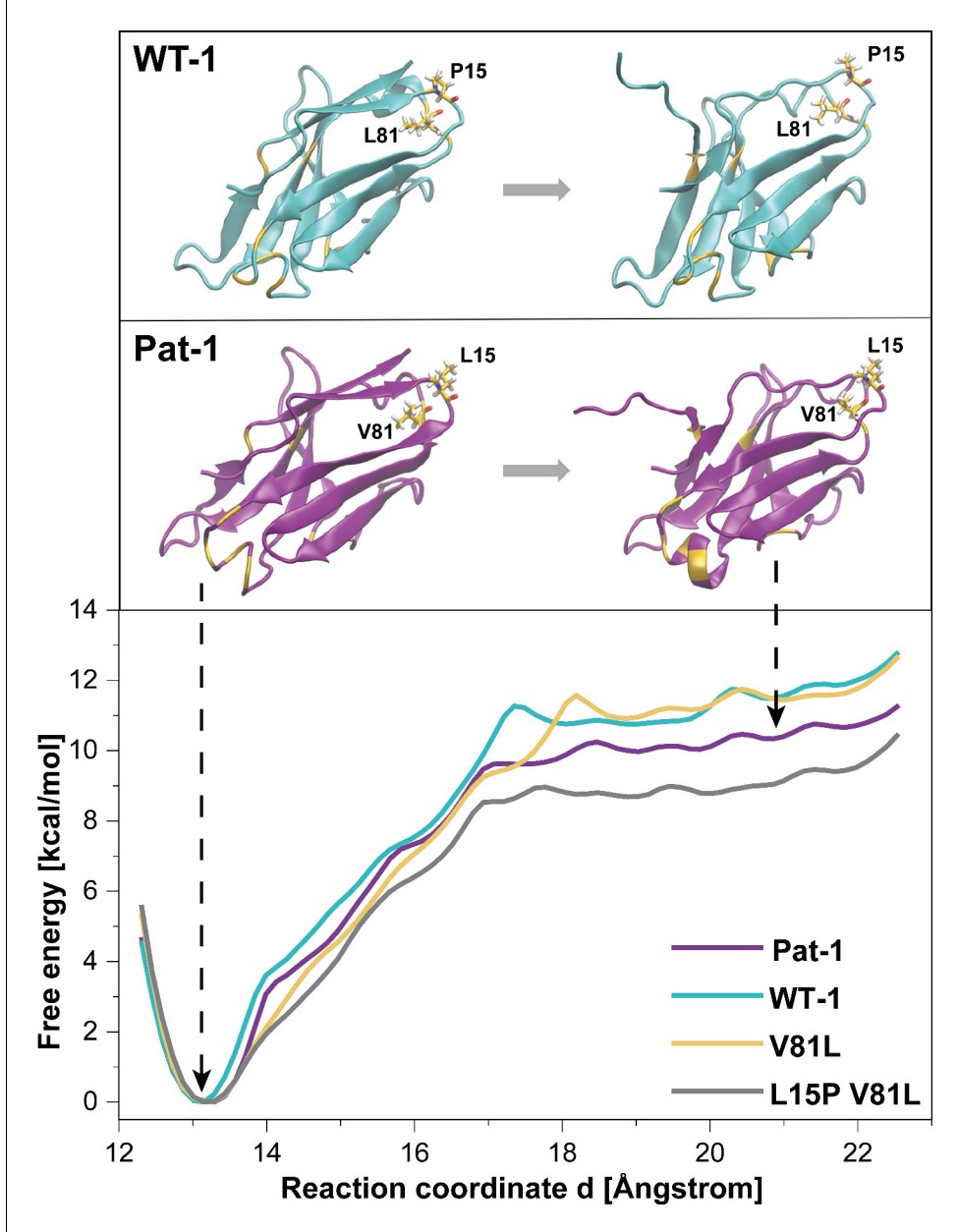

**Figure 7.** Potential of mean force (PMF) for the dissociation of the C-terminal segment. The free energy along the reaction coordinate was calculated by means of umbrella sampling simulations (see Materials and methods section). The reaction coordinate corresponds to the distance between the center of mass of the α-carbons of the C-terminal (residues 103 to 109) and the α-carbons of the rest of the protein (residues 1 to 102). A distance of ~13 Å indicates the position of the C-terminal segment as observed in the native structure whereas distances >~17.5 Å indicate dissociation of the C-terminal segment. Representative snapshots in the folded and unfolded states of WT-1 and Pat-1 are represented above the corresponding section of the PMF. The 11 resides which vary from WT-1 to Pat-1 are represented in yellow. The licorice representation corresponds to residues 81 and 15. Source files for the MD analysis are available in *Figure 7—source data 1*.

The online version of this article includes the following source data and figure supplement(s) for figure 7:

**Source data 1.** Reaction coordinate and free energy values from the MD simulation measurements.
**Figure supplement 1.** Root mean square deviation (RMSD) from corresponding time-averaged structures during MD Simulations.

resistant to amyloid formation. Only the Pat-1 $V_L$ readily formed fibrils. When viewed together, these findings suggest a strong effect of the LC context on the Pat-1 $V_L$ domain; the point mutations present in the $V_L$ domain only become effective when the $C_L$ domain has been removed (*Weber et al., 2018*). The current view holds that the overall destabilization of the $V_L$ domain is a major indicator for fibril formation propensity (*Blancas-Mejía et al., 2015*; *Morgan and Kelly, 2016*; *Nokwe et al., 2014*; *Ramirez-Alvarado, 2012*). The patient-derived mutant Pat-1 fits this general picture. However, a global destabilization alone is not sufficient to provide hints on the underlying processes leading to fibril formation. Furthermore, also $V_L$ domains with wildtype-like stability were found to form fibrils (*Nokwe et al., 2016*). For a mechanistic understanding of the disease, it is important to identify which of the sequence differences between germline and patient mutant are causative. The $V_L$ domains Pat-1 and WT-1 differ in 11 point mutations resulting in a general sequence identity of ~90%. By serially substituting each of the point mutations in Pat-1 with the corresponding amino acid present in WT-1, we identified one specific substitution, valine to leucine mutation at position 81, as the key modification responsible for reversing the pathogenic properties of Pat-1 and enhancing the overall stability of the $V_L$ domain. Amino acid frequencies in antibody sequences hint toward a negative effect of valine at position 81 (13 % compared to 77% for leucine) (*Johnson and Wu, 2000*; *Wu and Kabat, 1970*). That this substitution at position 81 is indeed sufficient to also render the germline $V_L$ domain amyloidogenic was demonstrated by introducing the V81 residue into the sequence of WT-1. The effect on the stability and fibril formation was very specific for the leucine residue, for example a substitution with the highly similar isoleucine residue did not yield comparable effects. Residue 81 was found to be part of a surface-exposed hydrophobic area in Pat-1 together with the two other mutated residues L15 and L82. Since surface-exposed hydrophobic areas are known to be energetically unfavorable (*Eisenhaber and Argos, 1996*; *Moelbert et al., 2004*; *Young et al., 1994*), we investigated these amino acids regarding their stability and amyloidogenic properties. Especially the L82Q substitution seemed to be a promising candidate since this mutation changed the nature of the amino acid side chain and of possible interactions. Besides, the amino acid frequency at this position decreases from 68% for glutamine to 1% for leucine in Pat-1, supporting the idea that this substitution could have a severe effect on domain architecture (*Wu and Kabat, 1970*). Against our expectations, the single L82Q substitution did not affect the stability of Pat-1 and the mutant still formed fibrils. However, a combination of L81 and Q82 increased the stabilizing effect observed for L81 and the variant remained soluble. This stresses the important role of the expanded hydrophobic surface area for amyloidogenesis that is reduced by the L82Q back mutation.

Surprisingly, substituting L15 with the germline residue P15 resulted in lower thermal and chemical stabilities than the ones observed for the patient mutant. The L15P mutation was also disadvantageous in combinations with L81 and/or Q82. All the P15-containing $V_L$ variants readily formed fibrils with an even earlier onset than the patient $V_L$ itself, stressing the negative effect of P15. This does not fit our expectations since according to the amino acid frequency a leucine at this position occurs in only 12% of the cases, while proline is present in the germline with a frequency of 61%.

Even though these results clearly show that stability changes are a major element affecting fibril formation, for a mechanistic explanation the structural consequences of the mutations need to be considered. In the crystal structure of Pat-1, V81 exhibits six hydrophobic interactions spread over FR2-4. These include interactions with the other two residues of the surface-exposed hydrophobic area, L15 and L82. Likewise, L81, present in WT-1, exhibits six hydrophobic interactions. However, the difference is that L81 reaches three amino acids in the C-terminal region (V109, V111 and L112), whereas V81 only extends its interactions to one residue (V111). This fits the analysis of the conformational dynamics of the two $V_L$s by HD/X which demonstrates a highly increased flexibility of the C-terminal segment in the patient-derived mutant as the underlying cause for destabilization and amyloid formation. A leucine at this position leads to a tighter packing of the domain than a valine residue. Therefore, neighboring residues are in closer proximity and form more hydrophobic interactions leading to a stabilization of the C-terminal region.

Changed hydrophobic interactions and thereby changed conformational dynamics can also explain the negative effect of P15. The leucine of Pat-1 at position 15 is tightly imbedded into a network of hydrophobic interactions. It facilitates five hydrophobic interactions to V81, L82 and A83 in FR3 and to V111 and L112 in the C-Terminal region. Even though these interactions increase the dynamics of the loop region around residue 15, the additional interactions to the C-terminal

segment positively affect the overall stabilization of the domain. A proline residue, on the other hand, induces a local change of the protein backbone structure and packing geometry abrogating hydrophobic interactions. This in turn affects a large segment of the protein although the $V_L$ variant still exhibits secondary, tertiary and quaternary structure comparable to the WT-1, indicating increased local dynamics. Thus, our results demonstrate a complex interplay of interactions that is responsible for the patient variant's pathogenicity. Our results further show that small and seemingly insignificant changes of a side chain can lead to a fatal change in intramolecular interactions, which are in the case of Pat-1 of hydrophobic nature.

The assays in this study were performed under simplified conditions in vitro compared to the environment the LCs experience in the human body. In the human body, many parameter including proteolysis, interactions with plasma factors and shear forces contribute to fibril formation and complicate the understanding of the pathogenicity. Furthermore, it is well-known that amyloid deposits consist not only of the disease-causing LC but also contain many other factors like glycosaminoglycans, lipids, apolipoprotein E or other proteins. Many of these factors have already been shown to influence fibril formation in the context of amyloid diseases (*Gellermann et al., 2005*; *Wyatt et al., 2012*). In the future their mechanistic influences on amyloid formation need to be determined to obtain a molecular understanding of disease progression, This may also offer new perspectives for potential treatment options.

In summary, our study presents a general strategy how to investigate LCs associated with AL amyloidosis. An important starting point is the determination of the patient LC sequence and the nature of the LC truncation present in the fibrils in vivo. The identification and classification of the mutated residues requires a comparison with the most homologous germline sequence and with the Kabat frequency database. This is the basis for a mutational analysis required to distinguish between active and silent mutations. The combination of structural and dynamic analyses of mutants together with fibril assays allows identifying the disease-causing residues and their contributions to stability and propensity for fibril formation. For defining the underlying molecular mechanism, analyses of the structural changes and dynamic consequences induced by the identified mutations are required. This information will contribute to better diagnosis and even treatment options.

# Materials and methods

**Key resources table**

| Reagent type (species) or resource | Designation | Source or reference | Identifiers | Additional information |
|---|---|---|---|---|
| Gene (*Homo sapiens*) | WT-1 | Uniprot IGLV2-14 | Uniprot: P01704 | |
| Gene (*Homo sapiens*) | Pat-1 | This paper | GenBank: MK962887 | |
| Strain, strain background (*E. coli*) | BL21-codon+ (DE3)-RIL | Stratagene | | |
| Strain, strain background (*E. coli*) | XL1blue | Stratagene | | |
| Biological sample (*Homo sapiens*) | abdominal fat tissue | University hospital Heidelberg | | |
| Recombinant DNA reagent | pet28b-WT-1 | invitrogen | | Point mutants obtained by site directed mutagenesis using NEBasechanger |
| Recombinant DNA reagent | pet28b-Pat-1 | invitrogen | | Point mutants obtained by site directed mutagenesis using NEBasechanger |

*Continued on next page*

*Continued*

| Reagent type (species) or resource | Designation | Source or reference | Identifiers | Additional information |
|---|---|---|---|---|
| Recombinant protein | Pat-1 LC, Pat-1 $V_L$, WT-1 LC, WT-1 $V_L$, Pat-1 $V_L$ and WT-1 $V_L$ point mutants | This paper | | Expression plasmids obtained from invitrogen (see: recombinant DNA reagent). Point mutant expression plasmids created using site directed mutagenesis (see: other). Protein purification according to Material and methods section |
| Commercial assay or kit | Wizard Plus SV Mini-Prep DNA purification | Promega | A1460 | |
| Software, algorithm | Origin 2018b | OriginLab Corporation | | |
| Software, algorithm | SedFit | Peter Schuck | | |
| Software, algorithm | Discovery Studio | BIOVA | | |
| Software, algorithm | Pymol | Schrödinger, DeLano Scientific LLC | | |
| Software, algorithm | Adobe Illustrator | Adobe Inc | | |
| Software, algorithm | PLGS and DynamX | Waters.com | | |
| Other | NEBaseChanger | NEB | https://nebasechanger.neb.com | Primer design for site directed mutagenesis |

Oligonucleotides were obtained from MW Biotech (Ebersberg, Germany). All chemicals were from Merck (Darmstadt, Germany) or Sigma (St. Louis, USA). All measurements were carried out in PBS buffer (10 mM $Na_2HPO_4 \times 2\ H_2O$; 1.8 mM $KH_2PO_4$; 2.7 mM KCl; 137 mM NaCl), pH 7.4 at 25°C, unless otherwise stated.

## Cloning of full length LC

The cDNA sequence of the light chain Pat-1 was obtained from CD138-enriched bone marrow plasma cells. The sequencing was performed as described elsewhere (*Annamalai et al., 2016*). The DNA sequence was deposited in the GenBank (https://www.ncbi.nlm.nih.gov/genbank/) under the accession number MK962887.

## Fibril extraction from fat tissue

To isolate fibrils from abdominal fat tissue, 25 mg patient tissue was diced with a scalpel and washed five times with 0.5 mL Tris calcium buffer (20 mM Tris, 138 mM NaCl, 2 mM $CaCl_2$, 0.1% (wt/vol) $NaN_3$, pH 8.0). Before each washing step, the sample was vortexed and centrifuged at 3,100 g for 1 min at 4°C. The supernatant was discarded. The pellet was dissolved in a solution of 5 mg/mL *Clostridium histolyticum* collagenase (Sigma) in Tris calcium buffer. The mixture was incubated overnight at 37°C and 750 rpm in a horizontal orbital shaker and thencentrifuged at 3100 g for 30 min at 4°C. The pellet was resuspended in 0.25 mL Tris ethylenediaminetetraacetic acid (EDTA) buffer (20 mM Tris, 140 mM NaCl, 10 mM EDTA, 0.1% (wt/vol) $NaN_3$, pH 8.0) and homogenized using a Kontes pellet pestle. The homogenate was centrifuged for 5 min at 3100 g at 4°C. The supernatant was removed and the homogenization step was repeated twice resulting in washing fractions 1–3. The remaining pellet was again homogenized with the pestle in 0.1 mL of ice-cold water and centrifuged for 5 min at 3100 g at 4°C. The supernatant was stored as water extract one and the step was repeated four more times resulting in extracts 2–5.

## Cloning, mutagenesis, expression and purification of LC/$V_L$ variants

DNA synthesis for the LCs and $V_L$s of Pat-1 and WT-1 was performed by Invitrogen (Carlsbad). Single mutations were introduced into the Pat-1 sequence by site-directed mutagenesis. Primers carrying the mutations were designed with NEBaseChanger and the PCR was performed according to the manufacturer's protocol. All variants were expressed and purified as previously described

(*Nokwe et al., 2014*; *Simpson et al., 2009*). In brief, the plasmids were transformed in *E. coli* BL21 (DE3)-star cells and protein expression took place at 37°C overnight. Cells were harvested and inclusion bodies were prepared as previously described (*Thies and Pirkl, 2000*). The pellet was solubilized and unfolded in 25 mM Tris-HCl (pH 8), 5 mM EDTA, 8 M urea and 2 mM β-mercaptoethanol at room temperature for a minimum of 2 hr. The solubilized protein was loaded onto a Q-Sepharose anion exchange column equilibrated in 25 mM Tris-HCl (pH 8.0), 5 mM EDTA and 5 M urea. The LCs and $V_L$s were eluted in the flow-through fractions and refolded by dialysis against 250 mM Tris-HCl (pH 8.0), 100 mM L-Arg, 5 mM EDTA, 1 mM oxidized glutathione and 0.5 mM reduced glutathione at 4°C overnight. To remove aggregates and impurities, the refolded proteins were purified using a Superdex 75 16/60 gel-filtration column (GE Healthcare, Uppsala, Sweden) equilibrated in PBS buffer. Recovery and purity of intact proteins were analyzed by SDS-PAGE.

## Crystallography

Initial crystallization hits for both WT-1 and Pat-1 were obtained using the vapor diffusion method at 20°C. Equal amounts of protein sample (about 30 mg\mL, solved in 20 mM Tris-HCl, pH 7.5, 40 mM NaCl) and reservoir solution were mixed for setting up 0.4 µL sitting drops. WT-1 protein crystallized in presence of 0.2 M CaCl$_2$, 0.1 M HEPES, pH 7.5, 28% PEG400. Requiring no further cryo-protection, crystals were directly vitrified in liquid nitrogen at 100 K. A high-resolution data set was measured at beam line ID30 at the European Synchrotron Radiation Facility (ESRF, Grenoble, France) using radiation of λ = 0.98 Å. For Pat-1, suitable crystals were grown in 2 µL hanging drops with reservoir solutions optimized around the initial crystallization condition of 0.5 M (NH$_4$)$_2$SO$_4$, 0.1 M trisodium-citrate, pH 5.6, 1.0 M Li$_2$SO$_4$. For all crystals, mother liquor supplemented with 40% glycerin was used for cryo-protection. To obtain experimental phasing information, crystals were soaked for 4 hr in drops containing potassium tetrachloroaureate-(III)-hydrate dissolved in mother liquor. An anomalous data set was measured at the peak wavelength of the Au (L-III) edge (λ = 1.039 Å, f'=−17.0, f''=10) at beam line X06SA at the Paul Scherrer Institute, Swiss Light Source (Villingen, Switzerland). In addition, a native data set was recorded at beam line ID30 at the ESFR. For all data sets, initial analysis, data processing, scaling and reduction were performed using the XDS software package (*Kabsch, 1993*). Further structure determination made use of different programs, distributed together by the ccp4i program suite (*Winn et al., 2011*). The WT-1 structure was solved to 1.55 Å resolution by Patterson search calculation techniques using PHASER (*McCoy et al., 2007*) and the atomic coordinates of PDB entry 4NKI (*Hao et al., 2015*) as a starting model. Applying the experimental phase information provided by the Au-SAD data set measured from the Pat-1 crystal, the automated structure solution pipeline Crank2 (*Skubák and Pannu, 2013*) generated an initial model to 3.5 Å resolution. This, in turn, proved to be suitable for solving the native Pat-1 data set, thereby expanding the phase information to 2.5 Å. Iterative rounds of model building and refinement with Coot (*Emsley et al., 2010*) and Refmac5 (*Vagin et al., 2004*) followed by addition of water molecules with applying the ARP/wARP software package (*Perrakis et al., 1997*) further improved the structure models for both WT-1 and Pat-1. This resulted in final R$_{values}$ of R$_{work}$ = 14.2% and R$_{free}$ = 16.5% or R$_{work}$ = 19.8% and R$_{free}$ = 24.5%, respectively. Besides, both models were found to have good stereochemistry as analyzed by Molprobity (*Chen et al., 2010*). Further details regarding data collection and refinement are listed in *Figure 2—source data 1*. Atomic coordinates and structure factors for WT-1 and Pat-1 have been deposited in the RCSB Protein Data Bank under the PDB IDs 6SM1 and 6SM2, respectively.

## Far-UV circular dichroism (CD) measurements

Thermal transitions were recorded using a Jasco J-715 spectropolarimeter (Jasco, Grossumstadt, Germany) equipped with a Peltier element. Protein unfolding was followed by monitoring the signal change at 205 nm at a heating rate of 30°/h. All measurements were performed using a 10 µM protein solution in a quartz cuvette with 1 mm pathlength.

## Fluorescence spectroscopy

Tryptophan fluorescence measurements were carried out in a 10 × 2 mm quartz cuvette using a FluoroMax-4 spectrofluorometer (Horiba Jobin Yvon, Bensheim, Germany). The measurements were performed with slit widths of 3 nm for excitation and 4 nm for emission, respectively. The protein

concentration was 1 µM and the temperature 20℃. Unfolding transitions were carried out by denaturing the samples overnight in GdmCl concentrations from 0 to 4 M. The fluorescence intensity was measured at 349 nm every second for 50 s, and the average was taken. Analysis of the data was carried out assuming a two-state unfolding as described previously (*Pace, 1986*; *Santoro and Bolen, 1988*).

## Analytical ultracentrifugation (AUC)

AUC measurements were carried out using a ProteomLab XL-I (Beckman, Krefeld, Germany) equipped with absorbance optics. The protein concentration for the measurements was 20 µM in PBS buffer. A total of 350 µL per sample was loaded into assembled cells with quartz windows and 12-mm-path-length charcoal-filled epon double-sector centerpieces. The measurements took place at 42,000 rpm in an eight-hole Beckman-Coulter AN50-ti rotor at 20℃. Sedimentation was continuously scanned with a radial resolution of 30 µm and monitored at 280 nm. Data analysis was carried out with SEDFIT using the continuous c(S) distribution mode (*Brown and Schuck, 2006*; *Schuck, 2000*).

## Thioflavin T (ThT) assay

ThT assays were performed in black 96 well microplates (#437112, Nunc, ThermoFisher Scientific, Roskilde, Denmark). The fibril formation kinetics were followed by measuring every plate well at 440 nm excitation and 480 nm emission wavelengths every 30 min with a Tecan Genios plate reader (Tecan Group Ltd., Männedorf, Switzerland) (*Gade Malmos et al., 2017*). To remove aggregates and oligomers and prevent seed formation during the assay, monomer isolation was performed prior to the experiment by ultracentrifugation in an Optima MAX-E ultracentrifuge, (Beckman, Krefeld, Germany). Assays were performed in a final volume of 250 µL per well with 20 µM protein 10 µM ThT in PBS buffer (pH 7.4) containing 0.5 mM SDS to support fibril formation (*Kihara et al., 2005*; *Nokwe et al., 2015*; *Yamamoto et al., 2004*) and 0.05% $NaN_3$. Microplates were covered with a Crystal Clear PP sealing foil (HJ-Bioanalytik GmbH, Erkelenz, Germany) and kept in the plate reader at 37℃ under continuous orbital shaking of 180 rpm.

## Transmission Electron Microscopy (TEM)

To obtain TEM micrographs, 10 µL samples were taken from the completed ThT-assay wells, applied onto a 200-mesh activated copper grid and incubated for 1 min. The samples were washed with 20 µL $H_2O$ and negatively stained with 8 µl of a 1.5% (w/v) uranyl acetate solution for 1 min. Excess solutions were removed using a filter paper. Micrographs were recorded on a JEOL JEM-1400 Plus transmission electron microscope (JEOL Germany GmbH, Freising, Germany) at 120 kV.

## Hydrogen/deuterium exchange-mass spectrometry (H/DX-MS)

H/DX-MS experiments were performed on a fully automated system equipped with a Leap robot (HTS PAL; Leap Technologies, NC), a Waters ACQUITY M-Class UPLC, a H/DX manager (Waters Corp., Milford, MA) and a Synapt G2-S mass spectrometer (Waters Corp., Milford, MA), as described elsewhere (*Zhang et al., 2014*). The protein samples were diluted in a ratio of 1:20 with deuterium oxide containing PBS buffer (pH 7.4) and incubated for 0 s, 10 s, 1 min, 10 min, 30 min or 2 hr. The exchange was stopped by diluting the labeled protein 1:1 in quenching buffer (200 mM $Na_2HPO_4 \times 2\ H_2O$, 200 mM $NaH_2PO_4 \times 2H_2O$, 250 mM Tris (2-carboxyethyl)phosphine, 3 M GdmCl, pH 2.2) at 1℃. Digestion was performed on-line using an immobilized Waters Enzymate BEH Pepsin Column (2.1 × 30 mm) at 20℃. Peptides were trapped and separated at 0℃ on a Waters AQUITY UPLC BEH C18 column (1.7 µm, 1.0 × 100 mm) by a $H_2O$ to acetonitrile gradient with both eluents containing 0.1% formic acid (v/v). Eluting peptides were subjected to the Synapt TOF mass spectrometer by electrospray ionization. Samples were pipetted by a LEAP autosampler (HTS PAL; Leap Technologies, NC). Data analysis was conducted with the Waters Protein Lynx Global Server PLGs (version 3.0.3) and DynamX (Version 3.0) software package.

## Molecular Dynamics (MD) simulations

Molecular dynamics (MD) simulations were performed employing the Amber16 simulation package (*Case et al., 2016*). In order to analyse the stability of the $V_L$ domain under specific mutations,

Umbrella sampling (US) simulations were carried out with the pmemd.cuda module of the Amber16 package. The simulated constructs were the wild type WT-1, the patient mutant Pat-1, the single back mutation of the patient mutant Pat-1 V81L and the double back mutation of the patient mutant Pat-1 V81L L15P. For this purpose, the Amber ff14SB force field and the TIP3P solvent model (*Jorgensen et al., 1983*) were used. Each construct was solvated in water in a periodic solvent octahedron box with a minimum distance of 10.0 Å between each atom of the protein and the edge of the periodic box. Apart from that, a neutralization of the solution was achieved by adding $Na^+$ and $Cl^-$ ions. Relaxation of the structure was carried out with an energy minimization of maximum 1500 minimization cycles. Each system was then heated up in steps of 100K, for 10ps each, until a temperature of 300 K was reached, whereby the system was harmonically restraint to the start structure with a restraint force constant of 25.0 kcal/molÅ$^2$. Afterwards, the system was equilibrated by gradually reducing the restraint force, in five steps of 10ps each, to a restraint force constant of 0.5 kcal/molÅ$^2$. For the heating and equilibration, MD-steps of 2 ps were used. Subsequently, hydrogen mass repartitioning (HMR) was performed in order to enable an increment in the simulation time step from 2 ps to 4 ps for the production simulations (*Hopkins et al., 2015*).

The US method allows the system to overcome an energy barrier by implementing an additional quadratic restraining potential to guide the system along a selected reaction coordinate. It possible to extract the free energy along the reaction coordinate of interest in which the configurations vary energetically and the system overcomes possible energy barrier. For the purpose of obtaining information about the folding stability of the protein, we simulated the system along a path in which the C-terminus dissociates from the protein. For this, we divided the dissociation path in 19 umbrella windows, which vary in the used harmonic restraining potentials, that is the restraining force constant (K) and the reference value around which the system is forced to stay close to (dref). These penalty potentials were then selected in such a manner that the distribution of states was shifted along the reaction coordinate, the distribution of states converged around the desired reference value for each umbrella window and allowing for sufficient overlap between neighboring distributions. The reaction coordinate was in this case the distance between the center of mass of the C-terminal (residues 103 to 109) and the remaining residues of the protein (1 to 102). The sets of umbrella windows generated were the following: i) 11 consecutive simulations with dref varying between 12.0 Å and 22.0 Å with a step of 1.0 Å and a force constant of K = 2.5 kcal / Å$^2$ mol, ii) eight consecutive simulations with dref varying between 14.0 Å and 17.5 Å with a step of 0.5 Å and a force constant of K = 4.0 kcal / Å$^2$ mol. Here, positional restraints with a force constant of Kpos = 0.05 kcal / Å$^2$ were applied to the alpha carbons (Cα) of residues 1 to 102. Each umbrella window was simulated for 100ns, but only for the last 50 ns of each window a trajectory was written and used to calculate the free energy profile (or potential-of-mean-force: PMF) by means of the WHAM algorithm (*Kumar et al., 1992*) along the reaction coordinate.

## Acknowledgements

We thank the patient and her family for their support of our scientific study. This work was supported by a grant from the DFG to the research unit FOR 2969. We thank Astrid König for her excellent support regarding protein crystallization. The staff of the beamline X06SA at the Paul Scherrer Institute, Swiss Light Source (Villingen, Switzerland) as well as the staff of the beamline ID30 at the European Synchrotron Radiation Facility (Grenoble, France) is acknowledged for help with data collection. Thanks to Florian Rührnößl for performing the HD/X measurements, to Quirin Emslander for his help with cloning the constructs, to Tuan Khac Nguyen and Clara Hipp for help with the biophysical experiments.

## Additional information

### Funding

| Funder | Grant reference number | Author |
|---|---|---|
| Deutsche Forschungsgemeinschaft | FOR 2969 | Johannes Buchner<br>Pamina Kazman<br>Marie-Theres Vielberg<br>María Daniela Pulido Cendales<br>Lioba Hunziger<br>Benedikt Weber<br>Ute Hegenbart<br>Martin Zacharias<br>Rolf Köhler<br>Stefan Schönland<br>Michael Groll |

The funders had no role in study design, data collection and interpretation, or the decision to submit the work for publication.

### Author contributions

Pamina Kazman, Conceptualization, Data curation, Formal analysis, Validation, Investigation, Visualization; Marie-Theres Vielberg, Michael Groll, Formal analysis, Investigation, Methodology; María Daniela Pulido Cendales, Software, Formal analysis, Investigation, Visualization; Lioba Hunziger, Data generation; Benedikt Weber, Conceptualization, Supervision; Ute Hegenbart, Stefan Schönland, Resources; Martin Zacharias, Formal analysis, Visualization; Rolf Köhler, Resources, Investigation; Johannes Buchner, Conceptualization, Resources, Data curation, Supervision, Funding acquisition, Project administration

### Author ORCIDs

Pamina Kazman (iD) https://orcid.org/0000-0003-2369-9726
Johannes Buchner (iD) https://orcid.org/0000-0003-1282-7737

### Decision letter and Author response

Decision letter https://doi.org/10.7554/eLife.52300.sa1
Author response https://doi.org/10.7554/eLife.52300.sa2

## Additional files

### Supplementary files

- Transparent reporting form

### Data availability

The primary DNA sequence data of this study have been deposited in GenBank (https://www.ncbi.nlm.nih.gov/genbank/) under the accession number MK962887. Atomic coordinates and structure factors for WT-1 and Pat-1 have been deposited in the RCSB Protein Data Bank under the PDB IDs 6SM1 and 6SM2, respectively. All data generated or analyses during this study are included in the manuscript and supporting files. Source data files have been provided for Figure 1—figure supplement 1, Figures 2, 5, 6 and 7.

The following datasets were generated:

| Author(s) | Year | Dataset title | Dataset URL | Database and Identifier |
|---|---|---|---|---|
| Koehler R | 2019 | *Homo sapiens* clone S3807 immunoglobulin lambda light chain variable region (IGL) mRNA, complete cds | https://www.ncbi.nlm.nih.gov/nuccore/MK962887 | NCBI GenBank, MK962887 |
| Kazman P, Vielberg | 2020 | Wild type immunoglobulin light | https://www.rcsb.org/ | RCSB Protein Data |

| | | | | | |
|---|---|---|---|---|---|
| MT, Cendales MDP, Hunziger L, Weber B, Hegenbart U, Zacharias M, Koehler R, Schoenland S, Groll M, Buchner J | | chain (WT-1) | | structure/6SM1 | Bank, 6SM1 |
| Kazman P, Vielberg MT, Cendales MDP, Hunziger L, Weber B, Hegenbart U, Zacharias M, Koehler R, Schoenland S, Groll M, Buchner J | 2020 | Mutant immunoglobulin light chain causing amyloidosis (Pat-1) | https://www.rcsb.org/structure/6SM2 | | RCSB Protein Data Bank, 6SM2 |

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
