## [Decision Letter]

**Acceptance summary:**

Systemic AL amyloidosis is caused by the aberrant proliferation of B cells that secrete high levels of immunoglobulin light chains (LCs) into a patient's circulation. The LCs are often aggregation prone, leading to sometimes lethal amyloid deposits in patient tissues. LCs are highly variable in sequence, and hence, it had been difficult to delineate the causative events of amyloid formation. This paper provides important insight into the processes that lead to amyloidosis by using a patient-derived LC sequence and determining precisely which mutations cause the protein aggregation thought to result in disease. By engineering LC variants carrying select mutations, they show that disease-linked changes in the protein sequence create a dynamic hydrophobic network that is more prone to aggregation. This work, therefore, helps us understand how mutations can destabilize protein folds and promote aggregation, and it provides a roadmap to identifying disease-causative mutations in complex diseases.

**Decision letter after peer review:**

Thank you for submitting your article "Fatal amyloid formation in a patient's antibody light chain is caused by a single point mutation" for consideration by *eLife*. Your article has been reviewed by three peer reviewers, one of whom is a member of our Board of Reviewing Editors, and the evaluation has been overseen by Huda Zoghbi as the Senior Editor. The following individual involved in review of your submission has agreed to reveal their identity: Ulrich Brinkmann (Reviewer #2).

The reviewers have discussed the reviews with one another and the Reviewing Editor has drafted this decision to help you prepare a revised submission.

Summary:

Systemic AL amyloidosis is caused by the aberrant proliferation of B cells that secrete high levels of immunoglobulin light chains (LC) into a patient's circulation. The LCs are often aggregation prone, leading to sometimes lethal amyloid deposits in patient tissues. LCs are highly variable in sequence, and hence, it had been difficult to delineate the causative events of amyloid formation.

In this manuscript, Kazman and coworkers investigated in detail the biophysical characteristics of an LC driving AL amyloidosis from a patient that shortly after initiation of chemotherapy had passed away. By comparing the patient sequence to LC databases, they conclude that the patient clone contained 11 mutations. Crystal structures, however, did not reveal any significant changes between the patient and wildtype LCs. Through mutations, they find a particular role of a single L81V mutation in destabilizing the LC fold and driving amyloid formation. Modeling and molecular dynamics simulations suggested that V81 is embedded in a hydrophobic network that is more dynamic in the patient than in the wildtype sequence. The higher dynamics in this hydrophobic surface might contribute to aggregation.

This manuscript is well done, address important issues relevant to aggregation diseases, and would be interesting for readers. This includes the careful analysis of mutations on stability and aggregation, including back mutation that could prevent, at least partially, amyloid formation. Their findings allows the authors to provide a reasonable rationale for how a critical mutation can drive disease.

There are no additional experiments that we request. However, we ask that the authors improve their discussion by including the issues raised below.

Essential revisions:

1) For many protein misfolding diseases, oligomers on the pathway to amyloid-fibril formation have been shown to be responsible for pathogenicity and not the fibrils per se. The assumption within the paper is that pathogenicity is directly to fibril formation. As a result of this, there are no studies attempting to identify/quantify oligomer formation. Some further discussion of this is desirable as it is different from much of the work done in the field of protein misfolding particularly with respect to neurodegeneration.

2) The ThT assays that assess fibril formation contained small amounts of SDS, a detergent certainly not present in patients. The reason for this addition was not stated. Please explain the reason for adding that reagent and confirm that presence of it does not interfere with the experimental results. The analysis of the ThT aggregation data is also qualitative rather than quantitative, i.e., mutations increase or decrease the propensity to form fibrils. Is there no more quantitative way of calculating propensity, similar to the calculation of t_1/2_ or lag time for sigmoidal kinetics observed in other systems?

3) Fibril formation in the patient occurs in an environment much more complex (dense) than in the experimental setting. In an ideal world, one would be able to assess under the same conditions, possibly in full serum. I realize that available technologies are not (yet?) compatible with such experimental approaches. The authors should, however, add a discussion of this topic including (potentially expected?) differences and similarities.

---

## [Author Response]

Essential revisions:1) For many protein misfolding diseases, oligomers on the pathway to amyloid-fibril formation have been shown to be responsible for pathogenicity and not the fibrils per se. The assumption within the paper is that pathogenicity is directly to fibril formation. As a result of this, there are no studies attempting to identify/quantify oligomer formation. Some further discussion of this is desirable as it is different from much of the work done in the field of protein misfolding particularly with respect to neurodegeneration.

Undoubtedly, the intermediates occurring on the way from monomer to amyloid are of high importance. Many researchers in this field assume that the oligomers are the actual toxic species. This is one major open question in the field of amyloidosis. In the case of AL, fibrils represent the end product of the pathway and are the species that can be detected in patient tissue and organs. It is assumed that they cause mechanical damage in the affected organs. Therefore, in this case, it is well justified to study fibril formation as it is tightly linked to pathogenicity. As suggested, we mentioned this aspect now in the Results section and we added a more detailed explanation in the Discussion section.

2) The ThT assays that assess fibril formation contained small amounts of SDS, a detergent certainly not present in patients. The reason for this addition was not stated. Please explain the reason for adding that reagent and confirm that presence of it does not interfere with the experimental results. The analysis of the ThT aggregation data is also qualitative rather than quantitative, i.e., mutations increase or decrease the propensity to form fibrils. Is there no more quantitative way of calculating propensity, similar to the calculation of t_1/2_ or lag time for sigmoidal kinetics observed in other systems?

We agree that the use of SDS needs further explanations. We thank the reviewer for pointing this out. The rationale is now mentioned in the Results section before the first fibril formation assay. We added citations from the literature on the use of SDS to accelerate fibril formation. Furthermore, we added another supplementary figure showing that SDS doesn’t change the initial native secondary structure, as proven by CD-spectroscopy.

As suggested, we also quantitated the changes during the kinetics of the fibril formation. We added a paragraph in the Results section and a supplementary figure on the quantification showing t_1/2_ and maximal ThT fluorescence values that allow a comparison between the different V_L_ variants.

3) Fibril formation in the patient occurs in an environment much more complex (dense) than in the experimental setting. In an ideal world, one would be able to assess under the same conditions, possibly in full serum. I realize that available technologies are not (yet?) compatible with such experimental approaches. The authors should, however, add a discussion of this topic including (potentially expected?) differences and similarities.

We thank the reviewer for raising this important aspect that should indeed be considered in the future. We added a paragraph on this issue, including supporting references, in the Discussion.